# Structure Features and Physicochemical Performances of Fe-Contained Clinoptilolites Obtained via the Aqueous Exchange of the Balanced Cations and Isomorphs Substitution of the Heulandite Skeletons for Electrocatalytic Activity of Oxygen Evolution Reaction and Adsorptive Performance of CO_2_

**DOI:** 10.3390/molecules28072889

**Published:** 2023-03-23

**Authors:** Chunlei Wan, Xueqing Cui, Ming Liu, Bang Xu, Jihong Sun, Shiyang Bai

**Affiliations:** Beijing Key Laboratory for Green Catalysis and Separation, Department of Chemistry and Chemical Engineering, Beijing University of Technology, Beijing 100124, China

**Keywords:** clinoptilolite, extra-framework Fe, fractal structures, isomorphous substitution, Fe species, adsorptive separations

## Abstract

Fe(III)-modified clinoptilolites (Fe-CPs) were prepared by hydrothermal treatment. The collapse of the heulandite skeletons was avoided by adjusting the pH value using HCl solution, showing the maximum relative crystallinity of the Fe-CPs at an optimal pH of 1.3. The competitive exchange performances between Fe^3+^ ions and H^+^ with Na^+^ (and K^+^) suggested that the exchange sites were more easily occupied by H^+^. Various characterizations verified that the hydrothermal treatments had a strong influence on the dispersion and morphology of the isolated and clustered Fe species. The high catalytic activity of the oxygen evolution reaction indicated the insertion of Fe^3+^ into the skeletons and the occurrences of isomorphic substitution. The fractal evolutions revealed that hydrothermal treatments with the increase of Fe content strongly affected the morphologies of Fe species with rough and disordered surfaces. Meanwhile, the Fe(III)-modified performances of the CPs were systematically investigated, showing that the maximum Fe-exchange capacity was up to 10.6 mg/g. Their thermodynamic parameters and kinetic performances suggested that the Fe(III)-modified procedures belonged to spontaneous, endothermic, and entropy-increasing behaviors. Finally, their adsorption capacities of CO_2_ at 273 and 298 K were preliminarily evaluated, showing high CO_2_ adsorption capacity (up to 1.67 mmol/g at 273 K).

## 1. Introduction

Clinoptilolite (CP), one of the abundant naturally occurring microporous zeolites, has a two-dimensional channel structure and is a member of the heulandite (HEU) group [1]. As described by Koyama and Galli et al. [2,3], CPs with two-dimensional micropore structures consist of channels A, B, and C formed by alternating 10- and 8-membered ring frameworks, respectively. Channels A (0.72 × 0.44 nm) and B (0.55 × 0.40 nm) are parallel to each other and to the c-axis of the unit cell, which intersects with a third channel (channel C = 0.41 × 0.40 nm) composed of an 8-membered ring [4]. Because of its unique micropore features and the physicochemical properties of the silicon-oxygen tetrahedrons and aluminum-oxygen tetrahedrons, CP presents a wide application potential in gas separation, environmental treatment, and industrial catalysis [5,6,7].

As is well known, the transition metal ions and their nanoparticles located in zeolites are effective modifiers of their properties. Fe-containing zeolites [8] currently attract extensive attention because of their advantages in the degradation of industrial organic dyes [8], selective catalytic reduction of nitrogen oxides [8,9], and selective adsorption of CO_2_ [10]. However, their preparation methods and treatment procedures have significant effects on the highly variable chemical performances of the iron ions in an environment of oxygen-containing ligands. The simultaneous existence of various types of Fe species, particularly this relationship, has not been unambiguously established.

The Fe(III) modification process in zeolites from Fe-containing salt solutions is not straightforward, which can be interpreted in the following two aspects: On one hand, the Fe(III) exchange process in silicon-rich zeolites is difficult because of the insufficient local negative framework charge to balance a bare trivalent cation. On the other hand, Fe^3+^ in relation to its source has a strong tendency to hydrolyze in aqueous solutions [11], which mainly exist as [Fe(H_2_O)_6_]^3+^, [Fe(H_2_O)_5_(OH)]^2+^, or [Fe(H_2_O)_4_(OH)_2_]^+^ single-ion aquo-complexes, as well as dimeric forms present as [Fe_2_(μ-O)(H_2_O)_8_]^4+^ oxo-bridged complexes, or [Fe_2_(μ-O)_2_(H_2_O)_8_]^4+^ di-oxo-bridged complexes. In high concentrations of non-acidified Fe(III) salt solutions, Fe^3+^ polymerization easily leads to formations of α-Fe_2_O_3_ or α-FeO(OH) [12].

In order to overcome these shortcomings, previous researchers have made much effort and proposed preparation strategies for achieving ion modification. For example, early researchers used ferrous salts as an iron source to solve the disadvantage of the insufficient local charge balances in the zeolite framework via a conventional ion exchange method. Feng and Hall [13] reported an aqueous exchange route using a saturated solution of FeC_2_O_4_ (about 10^−3^ M) in an inert gas. Base-exchange of the NaZSM-5 with Fe^2+^ allows up to 200% exchange to be achieved (calculated on the basis of 1Fe^2+^ = 2Na^+^). Prins and co-workers [14] further investigated the structures of the Fe species, and concluded that iron was precipitated onto the zeolite mainly as an FeC_2_O_4_ complex, which blocked the pores and impeded complete exchange of the cations. While most of these precipitates could be removed by extensive washing of the zeolite, the remaining iron species were transformed into iron oxide during subsequent thermal treatment. Obviously, it seems difficult to prepare Fe-ZSM-5 using an FeC_2_O_4_ aqueous solution. Chen and Sachtler [15] used FeSO_4_ as the ion-exchange precursor and found that the majority of the Fe^3+^ was present as an oxygen bridged binuclear iron complex, such as [(HO)Fe-O-Fe(OH)]^2+^. Bian and co-workers [16] demonstrated that the surfaces of the FeCl_2_-modified CPs were covered with grain-like clusters sized 100 nm. Despite using both Fe(II) salts and a range of anions in aerobic and anaerobic conditions, the extent of ion exchange was not significantly improved due to the high iron loadings. It is clear that these approaches are predominantly affecting impregnation rather than exchange.

In an attempt to prevent the formation of aggregated Fe-oxyhydroxide species, and therefore to enhance the extent of their exchange, Joyner and Stockenhuber [17] performed the preparations using carefully dried iron (III) nitrate dissolved in a dried methanolic solution. The results showed that only approximately 10% (exchange degree) was achieved, even after refluxing of the ZSM-5/methanol slurry. However, 80% of the cations could be exchanged via stirring at room temperature in combination with ultrasonic irradiation. Čapek and co-workers [18] further confirmed that the exchange process of the low concentrations of Fe ions in the FeCl_3_-acetylacetonate solution was effective in yielding Fe ions located predominantly in the cationic exchangeable sites of ZSM-5 frameworks. In contrast, Fe-zeolites containing both Fe ions in cationic exchangeable sites and oxo- or hydroxo-dinuclear Fe-species were obtained at higher Fe concentrations.

A new method for preparing ZSM-5 catalysts containing highly dispersed iron species was recently proposed by Nechita and co-workers [19], in which Fe(III) oxalate was used as the iron precursor, and an excellent iron dispersion inside the micropores of zeolite ZSM-5 was carried out via an aerobic aqueous exchange process in organic solvents. With respect to the Fe^3+^-containing CP, it is important to consider that the hydrated charge density of the Fe^3+^ cations is lower than that of Al^3+^ [20]. Therefore, possible dealumination and incorporation of Fe^3+^ cations within the zeolite lattice structure may occur during the cation exchange procedure. Isomorphous substitution of Fe^3+^ into the zeolite framework leads to acidity variations and morphology modifications, while the pore expansion caused by introducing Fe in the place of Al affects the product selectivity [21].

Based on the above-mentioned demonstrations, the purpose of this work is to propose a novel route for preparing Fe-modified CPs with a high content of Fe via Fe(III)-contained aqueous solution at low pH. By adding hydrochloric acid to the FeCl_3_·6H_2_O solution system, the pH value of the Fe-contained solutions was systematically adjusted from 2.0 to 0.9 for screening of a suitable modification process. Many experiments were carried out for establishing the relationships between the Fe^3+^content of the Fe-contained CPs and the modified parameters. The physicochemical and structural properties of resultant modified CPs were characterized via X-ray diffractive (XRD) and Small Angle X-ray Scattering (SAXS) patterns, Fourier transform infrared (FT-IR) and UV-vis spectra, thermal gravimetric (TG) behaviors, scanning electron microscopy (SEM) images, Electron Paramagnetic Resonance (EPR), electrochemical measurements (linear sweep voltammetry (LSV) and cyclic voltammetry (CV)), and N_2_ sorption-desorption isotherms. The thermodynamic parameters (enthalpy change (Δ*_r_H^θ^_m_*), entropy change (Δ*_r_S^θ^_m_*), and Gibbs energy change (Δ*_r_G^θ^_m_*)) and kinetic parameters (apparent activation energy (*E_a_*)) of the cation modification process through the Gibbs-Helmholtz equation and Arrhenius formula were calculated in order to understand the modification mechanism between Fe (III) cations and synthetic CP. Using various Fe-contained CPs as adsorbents, their adsorption capacities of CO_2_ at 273 and 298 K were preliminarily evaluated for possible applications in efficient use of CO_2_ storage.

## 2. Results and Discussion

### 2.1. Physicochemical Properties of Fe-Containing CPs

The XRD patterns of related samples and their corresponding calculated relative crystallinity were shown in Appendix A of the ESI section. The Fe(0.03) −H(0.05) −CP and Fe(0.06) −H(0.07) −CP with a higher relative crystallinity over 70% were selected to further explore the Fe^3+^ modified behaviors of the synthesized CPs. For comparison, Fe(0.03) −H(0.00) −CP and Fe(0.06) −H(0.00) −CP were prepared by the conventional ion modification method.

The XRD patterns of various CPs are depicted in Figure 1, Appendix A. As can be seen in Figure 1Aa, the diffractive peaks of CPs were found to be similar to that of conventional CP [22], suggesting the successful synthesis of CP. Meanwhile, (020), (200), and (131) peak intensities of NH_4_−CP (Figure 1Ab) became stronger than that of the parent CP (Figure 1Aa), but the intensities of other characteristic diffraction peaks remained nearly constant. Meanwhile, the XRD diffraction peaks ((020), (200) and (131)) of the NH_4_−CP shifted to a lower angle, as compared with that of the parent CP (Figure 1Aa), and the *d* spacing values of (020), (200) and (131) were expanded from 0.8909, 0.7892 and 0.3959 to 0.8937, 0.7907 and 0.3976 nm, respectively. Similar phenomena were also demonstrated by Li et al. [23], which may be caused by the larger diameter of NH_4_^+^.

However, the decrease in intensity of the diffraction peaks was observed for the contained-Fe CPs (as shown in Figure 1Ac,d, and Appendix A). As can be seen in Figure 1Ac,d, the diffractive peaks of the HEU structures, such as (020), (200), (111), (13-1), (131), (22-2), (42-2), (151), (62-1), and (061) were found almost missing, suggesting the structural collapse of Fe(0.03)−H(0.00)−CP−9 and Fe(0.06)−H(0.00)−CP−9 obtained via the conventional method. Particularly, their XRD patterns revealed the presence of a crystalline FeOOH phase [24] (β-FeOOH: JCPDS 34-1266) with a brown appearance. However, Fe(0.03)−H(0.05)−CP−9 and Fe(0.06)−H(0.07)−CP−9 (Figure 1Ae,f) showed patterns almost identical to that of the HEU structures with a cream color [22]. Meanwhile, no diffraction peaks corresponding to oxide crystallites (FeOOH phase) were observed, indicating good dispersion of the Fe species located on the HEU frameworks, though a slight decrease was observed in the intensities of the concerned peaks. As demonstrated by Alver and Sakizci [25], these phenomena may be partially caused by the relative acidity of the FeCl_3_ solutions, where the crystallinity decreased with increasing HCl concentrations. On the other hand, the isomorphic substitution of Fe^3+^ with Al^3+^ in the HEU framework led to less defined diffraction patterns due to its high charge to ionic radius ratio [20]. This likely indicated the variations in the atomic density and an increased disorder resulting in a lower crystallinity [26]. Rodríguez-Iznaga et al. [27] also found variation of the (020) peaks in the copper-silver bimetallic exchanged CP and believed that it was strongly linked to the extra-framework cations positioned in the mirror plane perpendicular to the *b*-axis in the HEU structures.

Figure 1B presents the UV-vis reflectance spectra of various Fe-containing CPs. As can be seen in Figure 1Be,f, Fe(0.03) −H(0.05) −9−CP and Fe(0.06) −H(0.07) −9−CP showed essentially two absorption bands located at ca. 210 and 272 nm, which were assigned to electronic transitions of the anion (e.g., O^2−^), that is, *t_2g_* and *e_g_* orbitals of Fe^3+^ within [FeO_4_]^−^ tetrahedra [9,28]. Obviously, these observations were associated with ligand to metal charge-transfer characteristics that involved the isolated four-coordinated and octahedral coordinated mononuclear Fe^3+^ [9]. The high-energy absorption band indicated that mononuclear Fe^3+^ should be dominant in these samples. However, as can be seen in Figure 1Bc,d, the bands appearing at 350 nm were assigned to octahedral Fe^3+^ as the low oligomeric clusters in Fe(0.03) −H(0.00) −9−CP and Fe(0.06) −H(0.00) −9−CP, while the bands appearing in the visible region (500 nm) were due to their more extensive clustering of octahedral Fe^3+^ [9]. For comparison, Figure 1Bc,d presented that Fe(0.03) −H(0.00) −9−CP and Fe(0.06) −H(0.00) −9−CP were brown in color, and their UV-vis spectra were composed mainly of a very broad line in the 200–700 nm range with a maximum absorption around 350 nm, belonging to a characteristic of iron oxide. Their bands centered at 510 nm corresponded to the *d-d* electron pair transition for Fe^3+^ in bulk Fe_2_O_3_ particles [9,29]. The absence of the peaks above 350 nm in Figure 1Ba,b suggested the absence of clustered Fe_2_O_3_ or Fe_3_O_4_ in CP and NH_4_−CP, which was dramatically different from Fe-containing CPs.

The SEM images of various CPs are shown in Figure 2. As can be seen in Figure 2a, the synthesized CP exhibited orderly stacked thin layer-like crystals with a smooth surface having a size of around 3.5 µm, which is found in good agreement with the reported literature [30]. In comparison, Fe(0.03)−H(0.05)−CP−3 (Figure 2b) and Fe(0.03)−H(0.05)−CP−9 (Figure 2c) showed almost the same morphologies with stacked lamellar structures in the size of around 3.0–3.5 µm. However, Figure 2c revealed that Fe(0.03)−H(0.05)−CP−9 had a certain degree of structure collapse, which may be caused by the acidic environment of the FeCl_3_ solutions with longer times, resulting in destruction of its HEU skeletons. In contrast, the morphology of Fe(0.03)−H(0.00)−CP−9 was quite different, as shown in Figure 2d, in which the rod-like clusters, about 150 nm long and about 60 nm in diameter, covered the surfaces of CP lamellas. These observations indicated that the acidic environment of the FeCl_3_ solutions had an obvious impact on the HEU structures and therefore the morphology of CPs.

Combined with the demonstrations of XRD pattern in Figure 1, the rod-like clusters dispersed in the lamellar surfaces of Fe(0.03)−H(0.00)−CP−9 (Figure 1Ac) belonged to β-FeOOH [24,31]. However, no similar clusters appeared on the surfaces of Fe(0.03)−H(0.05)−CP−9 (Figure 2c). Additionally, as shown in Appendix A, the individually colored elemental mapping images of CP, Fe(0.03)−H(0.00)−CP−9, and Fe(0.03)−H(0.05)−CP−9 showed that various elements such as O, Si, Al, and Fe were spread almost throughout the Fe-containing CPs.

Furthermore, the composition of CP before and after iron modification was analyzed by the energy dispersive X-ray (EDX) method. The typical EDX patterns for CP before and after the modification of Fe^3+^ were shown in Appendix A, and their corresponding amounts of Fe, O, Si, Al, Na, and K were collected in Appendix A. As can be seen in Appendix A, the characteristic signal of Fe^3+^ was barely observed in CP, while a clear signal of the presence of Fe^3+^ appeared in Fe(0.03)−H(0.05)−CP−9 and Fe(0.03)−H(0.05)−CP−9 (Appendix A).

The FT-IR spectra of various Fe-contained CPs are shown in Appendix A. As can be seen in Appendix A, Fe(0.03)−H(0.05)−CP−9 presented almost the same profiles as the synthetic CP (Appendix A), further indicating that the HEU structures of the resultant Fe-containing CPs remained intact even after Fe^3+^ modification. In detail, the peak located at 1638 cm^−1^ corresponded to deformation vibrations of adsorbed water [32]. The peak at 1400 cm^−1^ was due to NH_4_^+^ exchange in CP, which was related to the NH (primary amino) bending vibration [33]. The obvious peak located at 1046 cm^−1^, with a shoulder at 1210 cm^−1^, was associated with asymmetric stretching vibrations of the tetrahedral T-O-T (T = Si or Al) groups [34]. Others, centered at 610 and 465 cm^−1^, were assigned to the outer tetrahedral double ring and the bending vibration of the T-O (T = Si or Al) inside the tetrahedron, respectively, which is a key component of the primary structural units of CP [32,34]. The band at 796 cm^−1^ was assigned to the O-T-O stretching vibrations (T = Si or Al), which may be due to the presence of a quartz impurity in the samples [34]. The absorption peaks at approximately 1638 cm^−1^ were related to the deformation vibrations of H_2_O and stretching vibrations of -OH [35]. In accordance with works by Verdonck et al. [36], the prominent IR bands at 665 cm^−1^, as shown in Appendix A, can be assigned to Fe-O stretching vibrations or Fe-O-H bending vibrations. However, this peak was not observed in Fe(0.03) −H(0.05) −9−CP (Appendix A).

Obviously, the FT-IR spectra do not provide any more information about the coordination of Fe species, although we noticed that the asymmetric stretching band of the outer tetrahedron shifted from 1046 cm^−1^ in CP to 1055 cm^−1^ in Fe-containing CPs, due to the Fe introduction of the framework. Similar observations were also found by Rodriguez-fuentes et al. [37]. They elucidated that the introduction of Fe species in natural and modified CPs shifted this vibrational pattern toward higher frequencies.

Figure 3 present the TG profiles of CP and various Fe-containing CPs. As can be seen in Figure 3a, the weight loss process of CP could be divided into approximately three stages. In the temperature range from 373 to 473 K, the rapid mass loss (~4.5%) was documented by the steep slope of the TG curve, as it should be attributed to the loss of the water located in the CP cavities and bound to the non-framework cations [38]. Subsequently, the weight loss with a slight slope was less (~1.7%) during the temperature range from 473 to 673 K, corresponding to the desorption of chemisorbed water. Finally, its lighter slope in the temperature range from 673 to 1073 K was attributed to dehydroxylation [38].

As can be seen in Figure 3b,c, the weight losses of Fe(0.03)−H(0.05)−CP−3 and Fe(0.03)−H(0.05)−CP−9 were similar, but higher than that of the CP, which may be related to the Fe^3+^ radius and its charge intensity. Much of the H_2_O distributed in the structural cavities is strongly dependent on the extra-framework cations, and in particular, its hydration energy. As Alver et al. report [38], CPs containing high-hydration-energy cations, such as Ca^2+^ and Mg^2+^, could adsorb significantly more H_2_O than those such as Na^+^ and K^+^, while those with high-hydration energy cations also generally retain their H_2_O at higher temperatures. Interestingly, the weight losses of Fe(0.03)−H(0.05)−CP−3 and Fe(0.03)−H(0.05)−CP−9 (Figure 3b,c) were 5.53% and 5.35% at 373–473 K, respectively, showing a decreasing trend with increasing modification times. This phenomenon is due to the occurrences of the possible dealumination and partial isomorphs substitution of Fe^3+^ incorporated in the HEU skeletons, thus leading to the relatively hydrophobicity of Fe(0.03)−H(0.05)−CP−9. As reported by Sig et al. and Joshi et al. [21,39], increasing hydrophobicity of zeolite LTL appeared with increasing Fe^3+^ incorporation.

Figure 3d shows the TG curve corresponding to Fe(0.03)−H(0.00)−CP−9. Unlike other samples, Fe(0.03)−H(0.00)−CP−9 exhibited a rapid weight loss (~4.25%) at temperatures from 523 to 673 K, similar to that of bare FeOOH (Appendix A) [40], implying that Fe(0.03)−H(0.00)−CP−9 modified without an additive HCl solution may have a FeOOH phase, being consistent with the results of the SEM image (Figure 2d).

Figure 4 illustrates the ^29^Si-NMR profiles of Fe-containing CPs obtained at various modified conditions. As can be seen, the various CPs presented five different T sites with a significant overlap due to different T-T distances and T-O-T angles [41]. In which, the characteristic four resonance lines corresponded to the following assignment for 95 ppm Si (3 or 2Al), 100 ppm Si (2 or 1Al), 106 ppm Si (1 or 0Al), and 111 ppm Si(0Al) [37]. In detail, as can be seen in Figure 4A, the −95 ppm band was attributed to silicon with three or two aluminum neighbors Si (3 or 2Al) [37], while the broad lines at −95 ppm, as shown in Figure 4B–D, seemed to be more affected by the presence of paramagnetic iron in CP [37]. Similar observations were also reported by Rivera et al. [42], who found that partial extraction of Al and Fe occurred in the HEU skeletons after hydrochloric acid treatment of CP, triggering a decrease in the relative resonance intensity of −95 ppm of ^27^Al- and ^29^Si-NMR patterns. Particularly, the enhancement of −95 ppm relative resonance intensity of Fe(0.03)−H(0.05)−CP−9 in the ^29^Si-NMR pattern (Figure 4C) demonstrated that the Fe species recovered the Al initial position near the Si atoms. As reported by many researchers [43], these Fe species had almost same environments as the Al atoms in the HEU frameworks.

Additionally, four lines were deconvoluted after the acid treatment at pH = 1.3 (Figure 4B,C). The relative strength and linewidth of the −100 ppm line decreased significantly, as compared with that of CP (Figure 4A). This signal was associated with those Si bonded to 1 or 2Al, implying the extraction of Al from the frameworks. In the case of Fe(0.03)−H(0.05)−CP−3 and Fe(0.03)−H(0.05)−CP−9 (Figure 4B,C), the increase of the −105 and −111 ppm lines was at the expense of the −100 ppm. In general, the resulting dealumination mainly varied the environment around the Si(2Al) line, leading to generation of Si(1Al), possibly Si(0Al), losing of 1 or 2 Al species. Similarly, Fe(0.03) −HCl(0.00) −CP−9 (Figure 4D) modified in the FeCl_3_ solution presented the same phenomenon.

Therefore, the resonance at −95 ppm observed in the ^29^Si-NMR pattern of Fe(0.03)-HCl(0.05)−CP-9 (Figure 4C) and Fe(0.03)−H(0.00)−CP−9 (Figure 4D) revealed the contribution of Al and Fe atoms, suggesting the successful entry of Fe species into the HEU frameworks in the modification process, as compared with that of CP (Figure 4A).

Additionally, on the basis of Al-Yassir’s report [44] and Equation (2), the calculated Si/Al ratios were around 4.25 for CP, 4.47 for Fe(0.03)−H(0.00)−CP−9, 4.62 for Fe(0.03)−H(0.05)−CP−3, and even 4.75 for Fe(0.03)−H(0.05)−CP−9. Obviously, these observations suggested that both the increased modified times and the acid treatments should be beneficial for the removal of Al atoms from the frameworks.

EPR spectroscopy is an important tool for identifying the oxidation states of the transition elements in the aluminosilicate materials [37]. The above ^29^Si-NMR patterns establish the Fe species coordination in CPs. Figure 5 shows the EPR spectra acquired at 77 K for Fe-substituted CPs. As can be seen, all samples presented two signals: a sharp signal at g = 4.3 assignable to tetrahedral Fe^3+^ [28], and a broad signal at g = 2.0 assignable to octahedrally coordinated Fe^3+^ in the non-framework positions. Their deformation was the same as profiles reported by Bordiga et al. [45]. In detail, the signal intensity of Fe(0.03)−H(0.05)−CP−9 (Figure 5A) at g = 4.3 was greater than that of Fe(0.03)−H(0.05)−CP−3 (Figure 5B) and Fe(0.03) −HCl(0.00) −CP−9 (Figure 5C), suggesting that the increased modified times or the acid treatment were conducive to more coordinated tetrahedral Fe^3+^ entering the CP skeletons. On the contrary, the superimposed signals (g = 2.0) detected in Fe(0.03)−H(0.00)−CP−9 (Figure 5C) should be mainly attributed to β-FeOOH clusters when the hydrothermal modification of the CP was conducted using an FeCl_3_ solution without acid treatment, while a small portion of Fe species (g = 4.3) still had tetrahedral coordination to CPs framework.

However, UV-vis spectra of these Fe-contained CPs (as shown in Figure 1Be,f) indicated the creation of isolated octahedral Fe^3+^, which might be located in the balanced cation exchange positions of the extra-frameworks. Just from the EPR spectra, this information is difficult to derive, since the latter do not differ very much. These demonstrations illustrate that the position of the EPR signals alone cannot be used to draw conclusions about the coordination symmetry and the location in the CP matrix [46].

The N_2_ adsorption-desorption isotherms and the texture parameters of Fe-modified CPs were shown in Appendix A and collected in Appendix A. As can be seen in Appendix A, the adsorption capacity of the Fe-modified samples at very low relative pressures (*P*/*P*_0_ less than 0.1) proved the existences of micropores [47]. Meanwhile, the H3-type hysteresis appeared at *P*/*P*_0_ larger than 0.8, indicating the occurrence of aggregates or the appearance of slit-shaped pores with nonuniform size [48]. However, both micropore surface areas and micropore volumes of Fe(0.03)−H(0.05)−CP−3 and Fe(0.03)−H(0.05)−CP−9 were higher than that of CP; similarly, those of Fe(0.03)−H(0.05)−CP−9 were also higher than those of Fe(0.03)−H(0.05)−CP−3. These phenomena can be attributed to the removal of Na^+^ and K^+^ ions in CP after modification with acid solution, leading to the opening of microporous channels. Additionally, as can be seen in Appendix A, their mesopore size distributions obviously originated from inter-particle aggregations. As shown in Appendix A, the N_2_ adsorption-desorption isotherm of Fe(0.03)−H(0.00)−CP−9 presented an H4-type isotherm, demonstrating typical mesoporous characteristics of nanoparticles. The aggregation between β-FeOOH clusters and CP particles results in a wider mesopore size.

### 2.2. SAXS Patterns

The fractal structural evolutions of Fe-modified CPs with different modification times were elucidated by SAXS patterns. As shown in Figure 6A, the linearity of the *ln[I(q)]* versus *ln(q)* scattering profile drew the fractal property of the particles [49]. The straight lines in the *q* regions were determined to be in the range of 0.14 < *q* < 0.21 nm^−1^, which were fitted using the linear least-squares method. The corresponding slope continuously varied from −3.09 to −3.85. These results implied that all the CPs possessed surface fractal features [50]. In detail, the surface fractal dimension (*D_s_*) value gradually increased with the increasing of the modified times and Fe^3+^ content, from 2.044 for parent CP (Figure 6Aa), to 2.239 for Fe(0.03)−H(0.05)−CP−1 (Figure 6Ab), and 2.351 for Fe(0.03)−H(0.05)−CP−3 (Figure 6Ac), indicating that the surfaces of the Fe-containing CPs became disordered and rough [51]. Then, the *D_s_* values were no longer obviously changing with the increasing continuous content of Fe^3+^ (Figure 6Ad–f), indicating that the degree of disorder of CPs may have reached an upper limit.

The pair distance distribution function (PDDF) curves derived from SAXS data provided information about the shape and geometry features of the Fe-containing CPs. As shown in Figure 6B, the PDDF curves of various CPs lacked perfect symmetry, suggesting that the particles may present a flake-like morphology [52]. Furthermore, the maximum value of the intersection of the PDDF curves with the *X*-axis probably referred to the maximum size of the particles [53], showing the decreased tendencies from 51.8 nm for CP (Figure 6Ba) to 48.7 nm for Fe(0.03)−H(0.05)−CP−9 (Figure 6Bf). It may be because the acidic environment during the modified process had an effect on the lamellar thickness of CP [54]. Unfortunately, the particle sizes (over 300 nm) of CPs were well beyond the range of the assay. Therefore, the aforementioned results determined based on the PDDF data may not be suitable to the present work.

### 2.3. Thermodynamic and Kinetic Properties

Figure 7 and Appendix A profiled the influences of the modified parameters, such as temperature and time, on the performance of Fe(0.03)−H(0.05)−CP−x and Fe(0.06)−H(0.07)−CP−x using NH_4_-CP as the starting materials. As can be seen, the modified capacity of both Fe(0.03)−H(0.05)−CP−x and Fe(0.06)−H(0.07)−CP−x increased rapidly before 3 min and reached equilibrium roughly before 5 min. Similar phenomena also occurred when the modified temperature was increased from 298 to 333 K at the same modified times. Particularly, the modified capacity of Fe(0.03)−H(0.05)−CP−9 was 9.2 mg/g at 298 K, and increased to 10.6 mg/g at 333 K (as shown in Figure 7A–C). Rodríguez-Iznaga et al. [33] investigated the removal process of Mn^2+^, Co^2+^, and Ni^2+^ with NH_4_−CP from aqueous solutions. They also found that the modification process was mainly completed in the first few minutes, and was followed by a fast decay until equilibrium was reached. The increase in the modified capacity at higher temperatures may be due to the enhancement of the mass transfer driving force via increasing the modified temperature, which accelerated the cationic diffusion from aqueous solution to CP surfaces and promoted the accessibility of exchange sites [55].

However, Appendix A demonstrated that the modification procedure of Fe(0.06)−H(0.07)−CP−x needed more time to reach equilibrium than Fe(0.03)−H(0.05)−CP−x. The main reasons could be interpreted as follows. On one hand, the appearance of hydroxo-oligomers (Fe-OOH) at higher ionic strengths tends to block the micropores, making it difficult for Fe^3+^ to enter the microporous channels and achieving equilibrium [56]. On the other hand, the strong competition effect of H^+^ derived from the low pH environment also makes it difficult for Fe^3+^ to achieve equilibrium [30,55].

On the basis of the relationships between *ln k_a_* and (*RT*)^−1^ × 10^4^ at different modified times (Appendix A), the positive Δ*_r_H^θ^_m_* values for all samples indicated that the cationic modified process was endothermic (as shown in Appendix A), in good agreement with our previous investigations [23]. Therefore, the modified capacity at high temperature was larger than that at low temperature with the same modified times. Similarly, Barros et al. [57] observed that the exchange process of K^+^ with Zeolite NaA was endothermic at 303 and 315 K. They explained that the positive values of Δ*_r_H^θ^_m_* of the exchange process may be related to the dehydration of the hydrated in-going cations in achieving the most favorable exchange sites. Rodríguez-Iznaga et al. [33] found that Cu^2+^ originating from dehydrated [Cu(H_2_O)_6_]^2+^ easily reached the exchangeable sites during the exchange of Cu^2+^ with NH_4_-CP.

As can be seen in Appendix A, the calculated Δ*_r_G^θ^_m_* values for most samples were negative, indicating that the modification process was spontaneous. Although, their modification procedures gradually became more difficult due to their increased *E_a_* values with the enhanced modification times. Thus, we can speculate that the thermodynamic behavior of the Fe(III)-modification procedure in aqueous solution is not dominant. The Δ*_r_S^θ^_m_* values were calculated using the Gibbs-Helmholtz formula (as shown in mentioned above Equation (7)). As shown in Appendix A, the Fe^3+^ modified process belonged to an entropy enhancement. Similar results were also described by Pandey et al. [58]. However, the present results were not consistent with those reported by Argun et al. [59] and Tarasevich et al. [60]. Particularly, Argun et al. [59] found negative Δ*_r_S^θ^_m_* values during the removal of Ni^2+^ from aqueous solution using CP as an adsorbent, which was mainly a Ni^2+^ adsorption process. In contrast, our work investigated the Fe^3+^ modified behaviors with the exchangeable cations (Na^+^ and K^+^) of CP. Therefore, the purpose of washing with deionized water in the Fe^3+^ modified experiments was to remove the Fe^3+^ adsorbed on CP. As demonstrated by Barros et al. [57], the cation adsorption process was usually exothermic, whereas the cation exchange behavior could be endothermic and increase the entropic process. Herein, the contributions to Δ*_r_S^θ^_m_* may arise from modification between the Fe-containing aqueous solution and NH_4_−CPs [60], besides involving the variation in water-cation environments. Additionally, an overall negative or positive variation in entropy was also linked to the enthalpy values of the modified process.

Appendix A collects the calculated *E_a_* values of Fe(a) −H(b) −CP−x. As can be seen, the increased *E_a_* values implied that the modified process needed more energy and became more difficult with the enhanced modified times, which can be interpreted as follows. On the one hand, the Fe^3+^ first located the easily exchangeable sites at initial exchange stages. On the other hand, the concentration differences of Fe^3+^ distributed in between the CP phase and liquid phase decreased, leading to the decrease of their adsorption capacity and diffusion performance on the CP surface and its micropores. Meanwhile, the strong competition effect of H^+^ makes Fe^3+^ more difficult to arrive at exchangeable sites. The intra-particle diffusion coefficient also depended on the pH value in the solutions [30]. However, the *E_a_* values were around 7.4 to 22.9 kJ/mol, similar to Inglezakis’s results of 0.2–80 kJ/mol [61].

### 2.4. Electrocatalytic Performances of Fe-Containing CPs

The electrocatalytic performances of the Fe-containing CPs toward oxygen evolution reaction (OER) were tested in a 1.0 M KOH solution through a typical three-electrode system at a scan rate of 5 mV s^−1^, in which the assessment criterion for OER activity is generally an overpotential at a current density of 10 mA cm^−2^ [62]. The ohmic potential drop (iR) correction for the polarization curves was applied to all initial data, unless specifically stated [63].

Figure 8A presents the LSV curves of the Fe-containing CPs and representative control samples at a scan rate of 5 mV s ^−1^. As expected, bare CP (Figure 8Aa) displayed a poor electrochemical performance. However, Fe(0.03)−H(0.05)−CP−3 (Figure 8Ab) and Fe(0.03)−H(0.05)−CP−9 (Figure 8Ac) showed lower overpotentials at a current density of 10 mA cm^−2^. The probable reason may be due to that fact that Fe doping in CPs easily generates high-density vacancy defects with lattice distortion, leading to an increase in the number of exposed active sites, and therefore promoting the OER activity of Fe (0.03) −H(0.05) −CP. As shown in Figure 8Ad, the higher OER catalytic activity of the Fe (0.03) −H(0.00) −CP−9 is due to presence of a large number of FeOOH clusters on the surfaces of the modified CP. As Chen et al. demonstrates [64], the OER activity of FeOOH was conducive to the improvement of catalytic performance.

Generally, the larger the C_dl_ is, the more reaction sites are exposed on the catalyst surface and the higher the current density generated during the reaction. In order to directly evaluate the active site involved in the reaction, ECSA values of various CPs were represented by calculating C_dl_ through CV measurements at different sweep scan rates in a non-faradaic region. As can be seen in Figure 8B and Appendix A, the C_dl_ values were around 462.15, 896.35, 1100.00, and 1300.00 μF cm^−2^ for CP, Fe(0.03)−H(0.05)−CP−3, Fe(0.03)−H(0.05)−CP−9, and Fe(0.03)−H(0.00)−CP−9, respectively. The ECSA values of Fe(0.03)−H(0.05)−CP−9 were estimated to be about 32.50 cm^−2^, higher than that of CP (11.55 cm^−2^), Fe(0.03)−H(0.05)−CP−3 (22.41 cm^−2^), and Fe(0.03) −HCl(0.00) −CP−9 (27.5 cm^−2^). These results suggested that Fe(0.03)−H(0.05)−CP−9 possessed more active sites at the solid-liquid interfaces, which may be caused by the coordinated tetrahedral Fe^3+^ entering the CP skeletons.

Summaries of overpotentials (η) at 10 mA cm^−2^ and tafel slopes in 1.0 M KOH solution for OER properties obtained in this work and reported literature [65,66,67,68,69,70] was shown in Appendix A in the ESI section.

Two states of Fe cations are distributed in CP. One is octacoordinated Fe^3+^ appearing at the exchangeable sites, which is used to balance the aluminosilicate anions. The other is tetracoordinated Fe^3+^ located in the aluminosilicate skeletons, which is used in isomorphic substitute Al^3+^ positions. These results were demonstrated by ^29^Si-NMR patterns (as shown in Figure 4) and EPR spectra (as shown in Figure 5). As reported by Wu et al. [71] and Du et al. [65], the presence of octacoordinated Fe^3+^ facilitates the uplift in the O2p level and exhibits more oxygen vacancies than the tetracoordinated Fe^3+^, showing an effective low overpotential for OER. These vacancies further enhance the delocalization of the surrounding electrons, thus improving the conductivity of the Fe-CPs and their electron transfer performance [65]. Additionally, the octacoordinated Fe sites are more easily accessible to OER reactants, also facilitating their catalytic activity [72]. Gong et al. demonstrated that the tetracoordinated Fe located on the isomorphic substitution sites of the aluminosilicate skeletons in CP promotes the OER activity via the Lewis acid effect [73], which is beneficial to lowering the overpotential, and thus engendering more active sites in Fe-CPs.

Based on the above demonstrations, we proposed the synergy effects of tetracoordinated and octacoordinated-Fe on the principal OER active sites.

### 2.5. CO_2_ Adsorption Performance

The CO_2_ adsorption isotherms for all CPs at 273 and 298 K are shown in Figure 9. As can be seen, the CO_2_ uptake of CP and Fe(0.03)−H(0.05)−CP were much higher than Fe(0.03)−H(0.00)−CP within the examined pressure range. The CO_2_ adsorbed capacity of Fe(0.03)−H(0.05)−CP−3 within the examined relative pressure up to 1.0 reached 1.67 mmol·g^−1^ (Figure 9Ab) at 273 and 1.48 mmol·g^−1^ (Figure 9Bb) at 298 K. These values were higher than those of Fe(0.03)−H(0.05)−CP−9 (Figure 9Ac,Bc) and Fe(0.03)−H(0.00)−CP−9 (Figure 9Ad,Bd) at both 273 and 298 K, but lower those of parent CP (Figure 9Aa,Ba). Obviously, these results indicated that Fe-modified CPs did not improve the CO_2_ adsorbed capacity; similar finding was also reported by Moura et al. [74]. The probable reason may be because the introduction of Fe^3+^ promotes the formation of acidic sites and weakens the interaction with CO_2_.

The isosteric heats of CO_2_ adsorption for synthetic CPs are shown in Figure 10. As can be observed, the isosteric heats of CO_2_ adsorption on CP, Fe(0.03)−H(0.05)−CP−3, Fe(0.03)−H(0.05)−CP−9, and Fe(0.03)−H(0.00)−CP−9 were less than zero, indicating that the CO_2_ adsorption was an exothermic process [75]. Meanwhile, the calculated isosteric heat of adsorption was commonly less than 40 kJ·mol^−1^, meaning that physical adsorption dominates the CO_2_ adsorption behaviors on all CPs [28].

## 3. Materials and Methods

### 3.1. Materials

NaOH (99 wt%), KOH (85 wt%), Al(OH)_3_ (82 wt%), NH_4_Cl (99.5 wt%), and FeCl_3_·6H_2_O (99 wt%) were purchased from Tianjin Fuchen Chemical Reagents Co. (Tianjin, China). Ludox (1.2 g/mL, 30 wt% of SiO_2_) with an average particle size of 10–20 nm was provided by Qingdao Ocean Chemical Co. (Qingdao, China). Hydrochloric acid (HCl 37 wt%) was supplied by Beijing Chemical Factory (Beijing, China). Perfluorosulfonic acid-PTEE copolymer (Nafion solution, 5 wt%) was from Alfa Aesar, Qingdao, China. Acetylene black powder was from Guangdong Canrd New Energy Technology Co. (Dongguan, China). All chemicals were of analytical reagent grade. Deionized water with a resistivity of 18.25 MΩ·cm at 298 K was obtained by ZHIANG-Best Water Purifier in our lab.

### 3.2. Preparations

#### 3.2.1. Synthesis of CP

The preparation steps of CP included the following three stages: precursor preparation, hydrothermal synthesis, and post-treatment. First of all, certain amounts of NaOH, KOH, Al(OH)_3,_ and H_2_O (molar ratio: 1.38 Na_2_O:1.38 K_2_O:1 Al_2_O_3_:294 H_2_O) were mixed in a Teflon lined beaker and then stirred continuously at 150 °C for 3 h to obtain a clear aluminate-containing solution. Secondly, a mixture of 75 mL of deionized water and 44 mL of Ludox was poured into the prepared alumina sol, and then 10 wt% of natural CP, sieved with a 400-mesh sieve, was added and stirred continuously for 2 h. After that, the dilute precursor solution was transferred into Teflon-lined stainless-steel autoclaves and crystallized at 150 °C for 72 h. In the final step, the solid product was vacuum filtered and washed many times with large amounts of deionized water to ensure that the free ions were completely removed, and then in an oven at 100 °C for 12 h to obtain the synthesized CP.

#### 3.2.2. Ammonization of CP

The NH_4_^+^-exchanged CP with the mentioned-above CP was converted to single ions in 1 mol/L NH_4_Cl solution. The amount of CP was weighed and added to 200 mL of NH_4_Cl solution and stirred in an oil bath at 80 °C for 2 h. After filtering, it washed with copious amounts of deionized water and dried overnight, resulting in NH_4_-exchanged CP. The same procedure was repeated 7 times to obtain NH_4_−CPs with complete NH_4_^+^ exchange. More importantly, the complete exchange of Na^+^ and K^+^ with NH_4_^+^ was achieved by multiple washes, and the exchange capacity in Na^+^ and K^+^ was detected by Atomic Absorption Spectroscopy (AAS).

#### 3.2.3. Acid Resistance Test of CP

As demonstrated in the Introduction section, the presence of iron oxy-hydroxide species in an aqueous solution was considered to be unfavorable for exchange. An HCl solution was added into the Fe^3+^-containing modified system to inhibit the hydrolysis of iron ions, in which CP was treated in the solution at the ratio of 1 g:200 mL for 8 h. The obtained sample was denoted as Fe(x)−H(y)−CP (x and y represent the concentrations of FeCl_3_ and HCl solutions, respectively.)

The overall description of the modification protocol, including the concentrations of FeCl_3_ and HCl solutions, the modified temperatures, and the pH values of the system, and the obtained sample codes were listed in Appendix A (ESI) section.

#### 3.2.4. Ion Modification of CP in the FeCl_3_−HCl Solution

The Fe^3+^-containing and H^+^-containing aqueous solutions were prepared by dissolving FeCl_3_·6H_2_O and HCl in high-purity deionized water for the modification of NH_4_−CP. For example, during ion modification of Fe(0.03)−H(0.05)−CP, 1 g of dried NH_4_−CP was mixed separately with 200 mL of synthesized Fe^3+^ and H^+^ aqueous solutions, and stirred for 1, 3, 5, 7, 10, 15, and 20 min at 298, 313, and 333 K, respectively. The sample obtained for 20 min in the first modification process was used as a raw material for the second modification process. The same steps were repeated until the Fe^3+^ content in CP was no longer increasing. The resultant samples at 333K were labeled as Fe(x)−H(y)−CP−a (a = the modification times). In order to completely remove the residual chloride and salts adsorbed on the surface of the CPs, the obtained sample was vacuum filtered, washed exhaustively with a large amount of deionized water, and titrated with 0.1 mol/L AgNO_3_ solutions until no precipitation appeared in the filtrate. Finally, the obtained sample was then dried at 120 °C in air for 12 h.

Chemical analysis was performed by atomic absorption spectrometer (AAS) to determine the cation content in the sample with the following Formula (1)
(1)$$cationic\;exchange\;capacity\;\left(mg/g\right)=\frac{\rho_{cat}\times V_{s}}{\;\;\;m_{z}}$$
where *m_z_* refers to the mass of CP (g), *ρ_cat_* refers to the cation concentration in solution (mg/L), and *V_s_* refers to the volume of the solution (L). Therefore, the maximum Na^+^ (or K^+^, or NH_4_^+^) content of the CP was 55.20 (or 93.60, or 43.20) mg/g.

The above-mentioned schematic of the preparations was described as Appendix A in the ESI section.

### 3.3. Characterizations

XRD patterns were recorded using a Beijing Purknje General Instrument Corporation XD6 XRay diffractometer with Cu Kα radiation (λ = 1.54056 Å) in the range of 2*θ* = 5–50° for 4°·min^−1^ at 36 kV and 20 mA. The relative crystallinity of various CPs was calculated from the sum of intensities of ten main characteristic diffractive peaks, including (020), (200), (111), (13-1), (131), (22-2), (24-2), (151), (530), and (061), while the crystallinity of (Na, K)−CP was set to 1. The content of metal ions was recorded using Hitachi ZA3300 polarized AAS. The samples were completely dissolved in HNO_3_ solution at room temperature. In detail, 0.005 g of the obtained Fe-modified CPs was accurately weighed in a 10 mL plastic tube. For the accurate measurement of the mass of samples, AL 204 Electronic Balance (METTLER TOLEDO Co. Ltd., Columbus, OH, USA) was used, with a sensitivity of 0.0001 g. Then, a certain amount of HNO_3_ solution (volume fraction 2%) and 2–3 drops of HF aqueous solution (23 mol/L) were dropped at room temperature so as to completely dissolve it. Finally, this solution was transferred to a 10 mL volumetric flask. Diffuse reflectance ultraviolet-visible (UV-vis) spectra of Fe zeolites were recorded on a Shimadzu UV-2600 spectrometer equipped with an integrating sphere attachment, using BaSO_4_ as the reference. The morphologies were recorded using SEM (JEOL JEM-220) images with the microscope at 15.0 and 200 kV. IR-Prestige-21 FT-IR spectrum was used to record the samples in the wavenumber range of 400–4000 cm^−1^. The TG profiles were recorded using Perkin-Elmer Pyris Instruments TG-DSC thermal analyzer at a heating rate of 10 °C·min^−1^ and N_2_ flow rate of 20 mL·min^−1^. The EPR spectra of the samples were obtained at 77 K using a JEOL FA200 spectrometer, working at a modulation field of 100 kHz and equipped with an X band low temperature accessory for 77 K. The experimental conditions are as follows: Central field 400 mT; microwave power 1 mW; Average microwave frequency 9.45 GHz; exploration time 120 s. The *g* values were calculated from accurate measurements of magnetic fields and frequency parameters. ^29^Si-NMR spectra were measured via a Bruker AVANCE III 600 spectrometer at room temperature with a resonance frequency of 119.2 MHz and the chemical shift of ^29^Si was referenced to trimethylsilylation. The molar ratio of the framework Si/Al was calculated on the basis of the following Equation (2):(2)$$Si/Al=\frac{\sum_{n=0}^{4}A_{Si\left(nM\right)}}{\sum_{n=0}^{4}0.25nA_{Si\left(nM\right)}}$$

The adsorption-desorption isotherms of N_2_ were determined by JWGB jw-bk300 of Beijing Sci. & Tech. Co. Ltd., Shanghai, China. All samples were degassed for 6 h at a high vacuum at 120 °C, followed by N_2_ adsorption-desorption isotherms at liquid nitrogen temperature of 77 K. The surface area was calculated according to the BET equation.

All electrochemical tests were measured using a CHI 660E electrochemical workstation (Shanghai Chenhua Instrument Co., Ltd., Shanghai, China) in a three-electrode system with a 1.0 M KOH (pH = 14.0) electrolyte at room temperature.

### 3.4. SAXS Methods

The SAXS patterns were performed at the 1W2A station of the Beijing Synchrotron Radiation Facility by using synchrotron radiation as the X-ray source [76]. The incident X-ray wavelength was 0.154 nm. The sample-to-detector distance was 1600 mm and was calibrated by using the diffraction ring of a standard sample. The scattering vector magnitude *q* ranged from 0.09 to 3.04 nm^−1^ for the experiments. The detector readout noise (dark current) measured with a mask before the sample measurements was around 10 counts per second for the Mar165 CCD. The sample was loaded into a sample cell and sealed with Scotch tape on a groove. The thickness of the sample cell was approximately 1 mm. The scattering image was collected with an exposure time of 5 min by the single-frame mode with a “multi-read” of 2 times. The two-dimensional SAXS images were transferred to one-dimensional data by using the Fit2D software [53] (http://www.esrf.eu/computing/scientific/FIT2D (accessed on 25 January 2016)) and further processed with the S program package [76].

### 3.5. Thermodynamic and Kinetic Parameters

Accordingly, the thermodynamic equilibrium constants were evaluated on the basis of the following Equations (3)–(5) [55]:(3)$$k_{c}=\frac{A_{c}^{Z_{B}}\times {\left(1-A_{s}\right)}^{Z_{A}}}{{\left(1-A_{c}\right)}^{Z_{A}}\times A_{s}^{Z_{B}}}$$
where *k_c_* is the selective correction coefficient, and *A_s_* is the ratio of the concentration of Fe^3+^ ions in solution to the initial concentration of Fe^3+^ ions at equilibrium (*C*/*C_0_*). *A_c_* is the Fe^3+^ ion ratio of the amount of Fe^3+^ ion entering the phase of the used CP to the Fe^3+^ ion (*q*/*Q*), *q* is the content of cation in CP at ion Fe^3+^ equilibrium, *Q* is the maximum content of ammonia in CP. *Z_A_* and *Z_B_*, respectively, are the charges of the cations A and B, and the symbols *‘c’* and *‘s’* refer to the zeolite and solution phases, respectively.
(4)$$ln\;k_{a}=\left(Z_{B}-Z_{A}\right)+\int_{0}^{1}ln\;k_{c}dA_{c}$$
where *k_a_* is the equilibrium constant, *Z_A_* and *Z_B_* are the charges of cations *A* and *B*, respectively.
(5)$$ln\;k_{a}=\frac{\Delta_{r}H_{m}^{\theta }}{R}\times \frac{1}{T}+I$$

In thermodynamic functions (Gibbs-Helmholtz formula), *R* is Avogadro’s constant. *I* is a constant, and according to Equation (5), there is a linear relationship between ln *k_a_* and (*RT*)^−1^. Therefore, a fitting line can be obtained by plotting (*RT*)^−1^ with *ln k_a_*. Then based on the slope of line *k*, Δ*_r_H^θ^_m_* value can be calculated.

The Δ*_r_G^θ^_m_* value of modification process is calculated by the following Equation (6):(6)$$\Delta_{r}G_{m}^{\theta }=-\frac{RTln\;k_{a}}{Z_{B}\times Z_{A}}\;\;\;\;\;\;$$

Its Δ*_r_S^θ^_m_* value is calculated by the following Equation (7):(7)$$\Delta_{r}S_{m}^{\theta }=\frac{\Delta_{r}H_{m}^{\theta }-\Delta_{r}G_{m}^{\theta }}{T}$$

The kinetic parameters were calculated by the Arrhenius Equation (8) [77], as follows:(8)$$k=Ae^{-\frac{E_{a}}{RT}}$$
where *k* is the rate constant, *A* refers the frequency factor, *E_a_* denotes apparent activation energy, *R* and *T* are the gas constant and thermodynamic temperature, respectively.

### 3.6. Preparation of Electrodes and Electrochemical Measurements

Electrochemical catalyst ink was fabricated by dispersing 5 mg of samples (CP, Fe(0.03)−H(0.05)−CP−3, Fe(0.03)−H(0.05)−CP−9, and Fe(0.03)−H(0.00)−CP−9) and 2 mg acetylene black powder in a mixed solution containing Nafion solution (10 μL), deionized water (490 μL), and ethanol (1.5 mL), followed by ultrasonication for 1 h to generate a homogeneous suspension. Then, 20 μL of the above catalyst ink was uniformly cast onto a polished glassy carbon electrode (GCE, 5 mm) using a micropipette and dried at room temperature, which was then used as the working electrode with a loading amount of 0.5 mg cm^−2^.

The working electrodes were CP and Fe-containing CPs, the counter electrode was a graphite plate, and Ag/AgCl was used as the reference electrode. All the potentials were reported as one form of the reversible hydrogen electrode (RHE), calculated using the Nernst equation: E (RHE) = E (Ag/AgCl) + 0.197 + 0.059 pH) V. In the electrochemical study, LSV curves were obtained at a scan rate of 5 mV s^−1^, after the activation of 10 cycles of CV at a scan rate of 100 mV s^−1^. The electrochemically active surface areas (ECSAs) are usually measured by the double-layer capacitance (C_dl_) using CV at the non-faradaic potential range from 0.2 to 0.3 V on the basis of the equation: ECSA = C_dl_/C_s_, where C_s_ is corresponding to the specific capacitance (0.04 mF·cm^−2^). Ag/AgCl was used against the different scan rates (20, 40, 60, 80, and 100 mV s^−1^). Then C_dl_ were obtained by the slope of the fitted line.

### 3.7. Static Adsorption Measurements

Adsorption equilibrium isotherms for pure CO_2_ were measured using a JWGB (JW-BK 300, Beijing, China) at 273 and 298 K. Each sample was outgassed at 120 °C for 6 h before adsorption. After that, the measuring chamber was cooled down to the experiment temperature and the gas pressure (CO_2_) was increased stepwise until approximately 1.0 bar. The adsorption data for CO_2_ were fitted to the Langmuir Equation.

The experimental data of CO_2_ adsorption isotherms were fitted by the Freundlich-Langmuir (F-L) model, as shown in Equation (9), where it is assumed that the maximum adsorption on the surface of modified CPs relates to monolayer adsorption, and the adsorption (kinetics) energy remains a constant in the adsorption experiment [78].
(9)$$Q=\frac{q_{s}KC^{n}}{1+KC^{n}}\;$$
where *Q* is the moles adsorption capacity (mmol·g^−1^); *q_s_* is the saturation adsorption capacity (mmol·g^−1^). *K* and *n* are constants, and *C* is relative pressure.

To further verify the influences of the introduction of Fe^3+^ on the adsorption performance of CO_2_, the heats of adsorption of the Fe-modified CPs were calculated by using the Clausius-Clapeyron Equation (10) [79]:(10)$$ln\frac{P_{1}}{P_{2}}=\frac{\Delta H_{vap}}{R}\left(\frac{1}{T_{2}}-\frac{1}{T_{1}}\right)$$
where *P*_1_ and *P*_2_ represent the relative pressure under the temperatures of *T*_1_ and *T*_2_, Δ*H_vap_* represents isochoric adsorption heat of CO_2_, and *R* represents the gas constant (8.314 J·mol^−1^·K^−1^).

## 4. Conclusions

Fe-modified CPs were obtained by hydrothermal modification in a low pH acidic FeCl_3_ solution. Specially, the introduction of an HCl solution was beneficial to not only inhibit the hydrolysis of Fe^3+^, but also promote the insertion of Fe^3+^ into the CPs framework. The Fe-modification process was spontaneous and endothermic with increasing entropy. Various characterizations demonstrated that the octahedrally coordinated Fe species were distributed in the non-framework positions, while the tetrahedrally coordinated Fe^3+^ isomorph substituted Al^3+^ located in the HEU skeletons. The Fe-modified CPs presented satisfactory OER catalytic activity and CO_2_ adsorption performances, implying an excellent candidate for further explorations towards scale-up and potential applications.

## Figures and Tables

**Figure 1 molecules-28-02889-f001:**
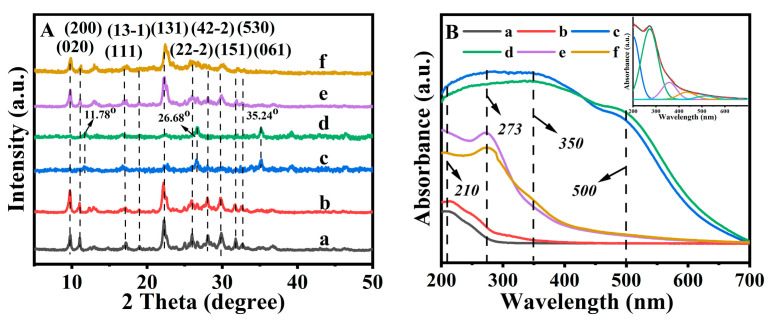
XRD patterns (**A**) and diffuse reflectance UV-vis spectra (**B**) of CP (a), NH_4_−CP (b), Fe(0.03) −H(0.00) −CP−9 (c), Fe(0.06) −H(0.00) −CP−9 (d), Fe(0.03) −H(0.05) −CP−9 (e), and Fe(0.06)−H(0.07)−CP−9 (f).

**Figure 2 molecules-28-02889-f002:**
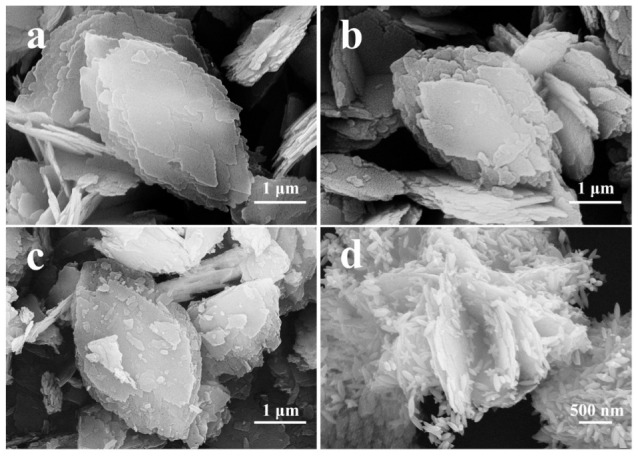
SEM images of CP (**a**), Fe(0.03)−H(0.05)−CP−3 (**b**), Fe(0.03)−H(0.05)−CP−9 (**c**), and Fe(0.03)−H(0.00)−CP−9 (**d**).

**Figure 3 molecules-28-02889-f003:**
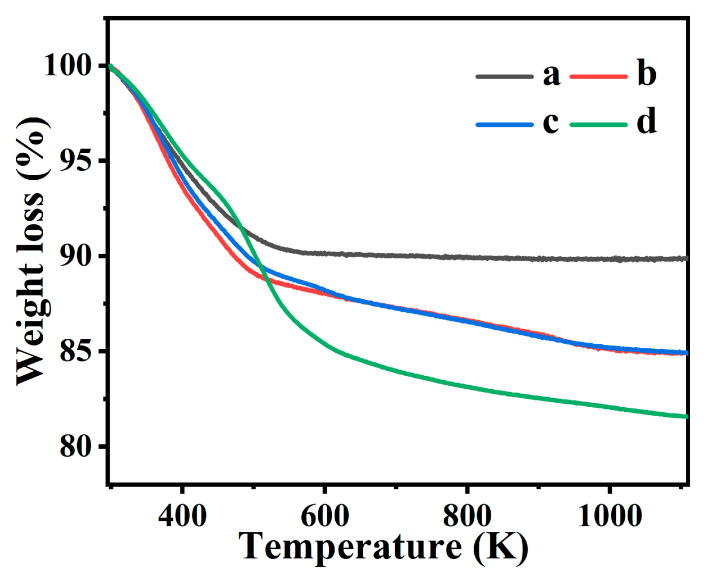
TG profiles of CP (**a**), Fe(0.03)−H(0.05)−CP−3 (**b**), Fe(0.03)−H(0.05)−CP−9 (**c**), and Fe(0.03)−H(0.00)−CP−9 (**d**).

**Figure 4 molecules-28-02889-f004:**
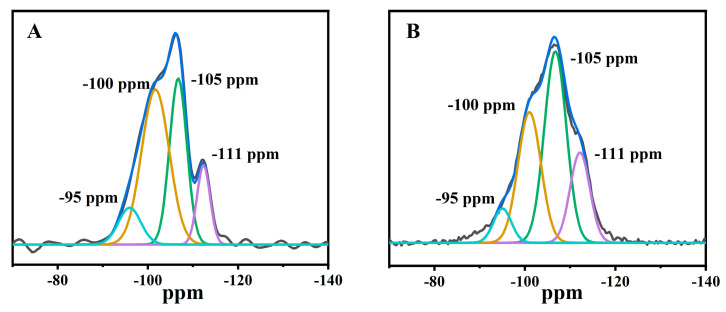
^29^Si-NMR patterns of CP (**A**), Fe(0.03) −H(0.05) −CP−3 (**B**), Fe(0.03) −H(0.05) −CP−9 (**C**), and Fe(0.03) −H(0.00) −CP−9 (**D**).

**Figure 5 molecules-28-02889-f005:**
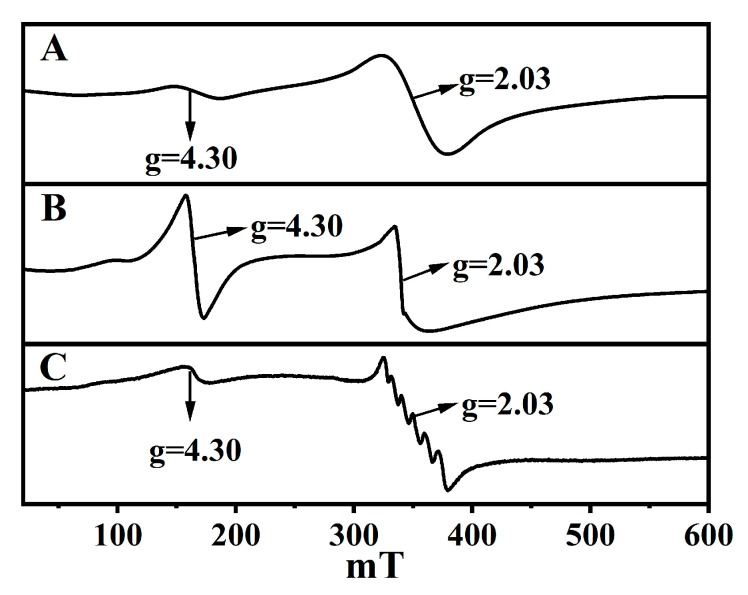
EPR spectra acquired at 77 K for Fe(0.03)−H(0.05)−CP−3 (**A**), Fe(0.03)−H(0.05)−CP−9 (**B**), and (**C**) Fe(0.03)−H(0.00)−CP−9.

**Figure 6 molecules-28-02889-f006:**
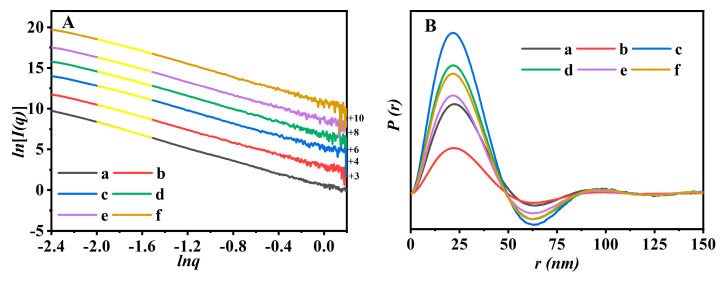
SAXS patterns (**A**) and their pair distance distribution function (PDDF) profiles (**B**) of CP (a), Fe(0.03)−H(0.05)−CP-1 (b), Fe(0.03)−H(0.05)−CP−3 (c), Fe(0.03)−H(0.05)−CP−5 (d), Fe(0.03)−H(0.05)−CP−7 (e), and Fe(0.03)−H(0.05)−CP−9 (f).

**Figure 7 molecules-28-02889-f007:**
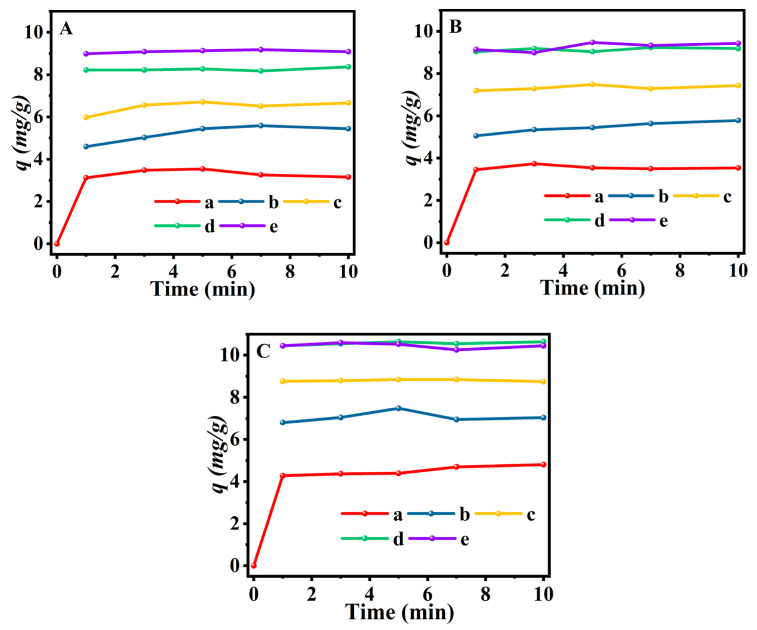
Kinetic curves of Fe(0.03)−H(0.05)−CP−x at 298 K (**A**), 313 K (**B**), and 333 K (**C**). x = 1 (a), 3 (b), 5 (c), 7 (d), and 9 (e).

**Figure 8 molecules-28-02889-f008:**
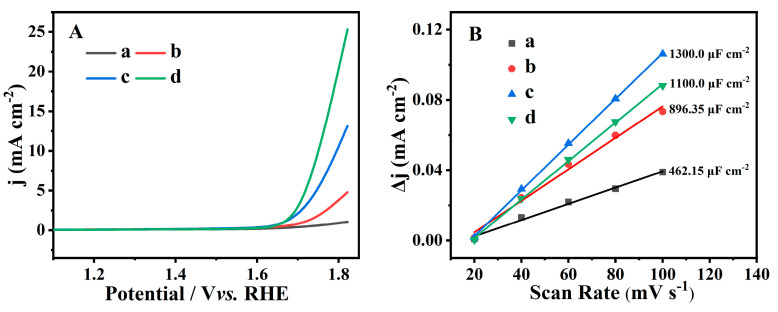
LSV curves (**A**) and plots of current density difference (Δj) against the scan rate (**B**) of CP (a), Fe(0.03)−H(0.05)−CP−3 (b), Fe(0.03)−H(0.05)−CP−9 (c), and Fe(0.03)−H(0.00)−CP−9 (d).

**Figure 9 molecules-28-02889-f009:**
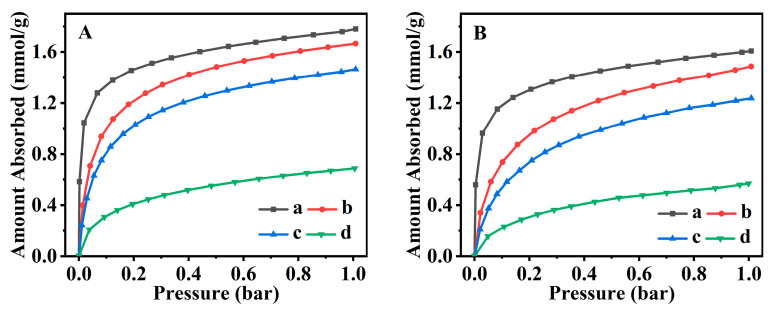
Equilibrium adsorbed isotherms of CO_2_ at 273 K (**A**) and 298 K (**B**) for CP (a), Fe(0.03)−H(0.05)−CP−3 (b), Fe(0.03)−H(0.05)−CP−9 (c), and Fe(0.03)−H(0.00)−CP−9 (d).

**Figure 10 molecules-28-02889-f010:**
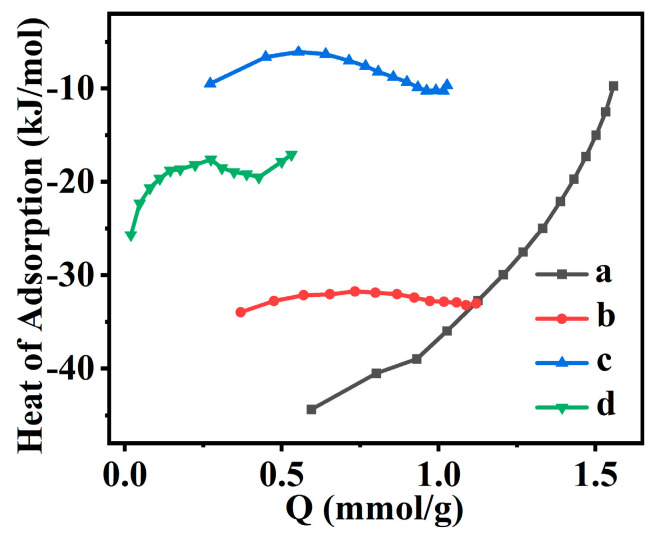
The isosteric heat of CO_2_ adsorption for CP (a), Fe(0.03)−H(0.05)−CP−3 (b), Fe(0.03)−H(0.05)−CP−9 (c), and Fe(0.03)−H(0.00)−CP−9 (d).

## Data Availability

Not applicable.

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
