# Peer review of "Structure Features and Physicochemical Performances of Fe-Contained Clinoptilolites Obtained via the Aqueous Exchange of the Balanced Cations and Isomorphs Substitution of the Heulandite Skeletons for Electrocatalytic Activity of Oxygen Evolution Reaction and Adsorptive Performance of CO2"

_molecules, 2023, doi:10.3390/molecules28072889_

Round 1
Reviewer 1 Report
Manuscript ID:molecules-2145851: “Structure features and physicochemical performances of Fe-contained clinoptilolites obtained via the aqueous exchange of the balanced cations and isomorphs substitution of the heulandite skeletons for electrocatalytic activity of oxygen evolution reaction"
Comment #1: The authors claimed the synergic effects. However, for this point, more explanations should be provided to shed light on this.
Comment #2: The manuscript should have an electrochemical impedance (EIS) and tafel plots.
Comment #3: For stability in OER it is necessary to carry out stability studies, using Cyclic Voltammetry and amperometry (10 mA cm-2)
Comment #4: As there are many oxygen evolution reaction papers published in the literature, the performances of the composites prepared in this study should be compared with the results reported in the literature.
Author Response
Response Reviewer 1
Reviewer 1#:
Response 1#: We thank the reviewer for the useful and positive comments. The manuscript was seriously revised according to your constructive suggestions.
Comment 1-01: The authors claimed the synergic effects. However, for this point, more explanations should be provided to shed light on this.
Response 1-01: We thank the reviewer’s useful suggestions.
Two states of Fe cations are distributed in CP, one is octacoordinated Fe3+ appeared at the exchangeable sites, which is used to balance the aluminosilicate anions, and the other is tetracoordinated Fe3+ located in the aluminosilicate skeletons, which is used to isomorphic substitute Al3+ positions. These results were demonstrated by 29Si-NMR patterns (as shown in Figure 4) and EPR spectra (as shown in Figure 5).
As reported by Wu et al. [1] and Du et al. [2], the presence of the octacoordinated Fe3+ is conductive to facilitate the uplift in the O2p level and exhibits more oxygen vacancies than the tetracoordinated Fe3+, showing an effective low overpotential for O2 evolution reaction (OER). While, these vacancies further enhance the delocalization of the surrounding electrons, thus improving the conductivity of the Fe-CPs and their electron transfer performances [2]. Additionally, the octacoordinated Fe sites are more easily accessible to OER reactants, also facilitating their catalytic activity [3].
Gong et al. also demonstrated that the tetracoordinated Fe located on the isomorphic substitution sites of the aluminosilicate skeletons in CP promotes the OER activity via Lewis acid effect [4], which is beneficial to lowering the overpotential, and thus engendering more active sites in Fe-CPs.
Based on the above demonstrations, we proposed the synergy effects of the tetracoordinated- and the octacoordinated- Fe on the principal OER active sites.
Appendix references:
[1] Wu, T.; Sun, S.; Song, J. Iron-facilitated dynamic active-site generation on spinel CoAl2O4 with self-termination of surface reconstruction for water oxidation. Nature Catalysis 2019, 2, 763-772.
[2] Du, Q.; Su, P.; Cao, Z. Construction of N and Fe co-doped CoO/CoxN interface for excellent OER performance. Catal. Sci. Technol. 2021, 29, 00293.
[3] Bao, J.; Zhang, X.; Fan, B.; Zhang, J.; Zhou, M.; Yang, W.; Hu, X.; Wang, H.; Pan, B.; Xie, Y. Ultrathin Spinel-Structured Nanosheets Rich in Oxygen Deficiencies for Enhanced Electrocatalytic Water Oxidation. Angew. Chem. 2015, 127, 7507-7512.
[4] Gong, L.; Chng, X.; Du, Y.; Xi, S.; Yeo, B. Enhanced Catalysis of the Electrochemical Oxygen Evolution Reaction by Iron(III) Ions Adsorbed on Amorphous Cobalt Oxide. ACS Catalysis 2017, 8, 807-814.
The mentioned-above demonstrations were added on Pages 16-17 Lines 646-662 in the revised manuscript, as follow:
Two states of Fe cations are distributed in CP, one is octacoordinated Fe3+ appeared at the exchangeable sites, which is used to balance the aluminosilicate anions, and the other is tetracoordinated Fe3+ located in the aluminosilicate skeletons, which is used to isomorphic substitute Al3+ positions. These results were demonstrated by 29Si-NMR patterns (as shown in Figure 4) and EPR spectra (as shown in Figure 5). As reported by Wu et al. [69] and Du et al. [70], the presence of the octacoordinated Fe3+ is conductive to facilitate the uplift in the O2p level and exhibits more oxygen vacancies than the tetracoordinated Fe3+, showing an effective low overpotential for OER. While, these vacancies further enhance the delocalization of the surrounding electrons, thus improving the conductivity of the Fe-CPs and their electron transfer performances [70]. Additionally, the octacoordinated Fe sites are more easily accessible to OER reactants, also facilitating their catalytic activity [71]. Gong et al. also demonstrated that the tetracoordinated Fe located on the isomorphic substitution sites of the aluminosilicate skeletons in CP promotes the OER activity via Lewis acid effect [72], which is beneficial to lowering the overpotential, and thus engendering more active sites in Fe-CPs.
Based on the above demonstrations, we proposed the synergy effects of the tetracoordinated- and the octacoordinated-Fe on the principal OER active sites.
Comment 1-02: The manuscript should have an electrochemical impedance (EIS) and tafel plots.
Comment 1-03: For stability in OER it is necessary to carry out stability studies, using Cyclic Voltammetry and amperometry (10 mA cm-2).
Response 1-02 and 03:
Thank you very much for your helpful suggestion.
However, it is indeed difficult for us to timely perform the above experiments, since it is the Chinese Spring Festival holiday and the limited 10 days for revised manuscript. Hopefully, the reviewers can understand.
But, the electrochemical impedance properties and their cyclic stability of the voltammetry and amperometry will be further investigated in the following work.
Comment 1-04: As there are many oxygen evolution reaction papers published in the literature, the performances of the composites prepared in this study should be compared with the results reported in the literature.
Response 1-04: Thank you very much for your advice.
Based Reviewer’s suggestion, the reported catalytic results for the OER performances were summarized, as shown in Table 1.
Table 1. Summaries of overpotentials (η) at 10 mA cm-2 and tafel slopes in 1.0 M KOH solution for OER properties obtained in this work and reported literature.
catalysts |
η (mV) |
Tafel slope (mV dec-1) |
Reference |
Fe(0.03)-H(0.05)-CP-9 |
560 |
129 |
this work |
Fe(0.03)-H(0.00)-CP-9 |
510 |
79 |
this work |
commercial RuO2 |
330 |
76.3 |
[1] |
Co0.89Fe0.11O-N |
304 |
52.7 |
[1] |
MIL-53(Fe) |
233 |
88.7 |
[2] |
CoNiFe ZIF-NFs |
273 |
87 |
[3] |
Fe-Co-CN/rGO |
308 |
138 |
[4] |
FeOxCF-8 |
408 |
93 |
[5] |
CoFe2O4/biocarbon |
417 |
-- |
[6] |
Appendix references:
[1] Du, Q.; Su, P.; Cao, Z.; Yang, J.; Price, C.; Liu, J. Construction of N and Fe co-doped CoO/CoxN interface for excellent OER performance. Catal. Sci. Technol. 2021, 29, e00293.
[2] Nivetha, R.; Kollu, P.; Chandar, K.; Pitchaimuthu, S.; Jeong, S.; Grace, A. Role of MIL-53(Fe)/hydrated–dehydrated MOF catalyst for electrochemical hydrogen evolution reaction (HER) in alkaline medium and photocatalysis. RSC advances 2019, 9, 3215-3223.
[3] Sankar S.; Manjula K.; Keerthana G.; Babu B.; Kundu S. Highly stable trimetallic (Co, Ni, and Fe) zeolite imidazolate framework microfibers: an excellent electrocatalyst for water oxidation. Cryst. Growth Des. 2021, 21, 1800-1809.
[4] Fang W.; Wang J.; Hu Y.; Cui Q.; Zhu R.; Zhang Y.; Yue C.; Dang J.; Cui W.; Zhao H.; Li Z. Metal-organic framework derived Fe-Co-CN/reduced graphene oxide for efficient HER and OER. Electrochim. Acta. 2021, 365, 137384.
[5] Yan F.; Zhu C.; Wang S.; Zhao Y.; Zhang X.; Chen Y. Electrochemically activated-iron oxide nanosheet arrays on carbon fiber cloth as a three-dimensional self-supported electrode for efficient water oxidation. J. Mater. Chem. A. 2016, 4, 6048-6055.
[6] Liu S.; Bian W.; Yang Z.; Tian J.; Jin C.; Shen M.; Zhou Z.; Yang R. A facile synthesis of CoFe2O4/biocarbon nanocomposites as efficient bi-functional electrocatalysts for the oxygen reduction and oxygen evolution reaction. J. Mater. Chem. A. 2014, 2, 18012-18017.
The mentioned-above demonstrations were added on Page 16 Lines 643-645 in the revised manuscript, as follow:
Summaries of overpotentials (η) at 10 mA cm-2 and tafel slopes in 1.0 M KOH solution for OER properties obtained in this work and reported literature was shown in Table S5 in the ESI section.
Reviewer 2 Report
The article discusses oxygen evolution’s electrocatalytic activity and adsorptive properties of CO2. The authors analyzed these properties with Fe-contained clinoptilolites, examining the structure and physicochemical performance of such zeolites. These zeolites were created via cation exchange as well as isomorph substitution. This report could be considered for publication after addressing a few comments.
11) A schematic of the preparations would be great to add.
22) EDX could be used to further analyze the zeolite and its composition.
Author Response
Response to Reviewer 2
Please check the attachment.

Round 2
Reviewer 1 Report
Manuscript ID:molecules-2145851: “Structure features and physicochemical performances of Fe-contained clinoptilolites obtained via the aqueous exchange of the balanced cations and isomorphs substitution of the heulandite skeletons for electrocatalytic activity of oxygen evolution reaction"
I am satisfied with the modifications made with the authors, I accept the manuscript for publication.